# Implementation of the My Abilities First Tool: A qualitative study on the perceptions of professionals, caregivers, children, and adolescents with disabilities

Roselene F. Alencar[1]*, Egmar Longo[2], Verónica Schiariti[3], Caline C. A. F. Jesus[4], Rocío P. Carrión[5], Carolina D. L. Alvarez[1], Monique L. G. Coelho[6], Tatiana F. C. Pereira[7], Maria C. L. Cruz[1], Sanzia S. V. Melgão[1], Ana Raquel R. Lindquist[1]

1 Department of Physiotherapy, Federal University of Rio Grande do Norte, Natal, Rio Grande do Norte, Brazil, 2 Department of Physiotherapy, Federal University of Paraíba, João Pessoa, Paraíba, Brazil, 3 Division of Medical Sciences, University of Victoria, Victoria, BC, Canada, 4 Postgraduate Program in Decision Models and Health, Federal University of Paraíba, João Pessoa, Paraíba, Brazil, 5 Faculty of Physiotherapy, Hemi Child-Research Unit, University of Castilla-La Mancha, Spain, 6 Postgraduate Program in Public Health Collective Health Center Health Sciences Center, Rio Grande do Norte, Brazil, 7 Edmond and Lily Safra International Institute of Neurosciences, Santos Dumont Institute, Postgraduate Program in Neuroengineering, Macaiba, Brazil

* roselene.alencar@ufrn.br

## Abstract

### Objective

To analyze the perceptions of professionals, caregivers, children, and adolescents with disabilities regarding the implementation of the My Abilities First (MAF) tool in Specialized Child Rehabilitation Centers (CERs).

### Method

This is a qualitative research based on Reflexive Thematic Analysis (RTA). The study involved twenty-seven intentionally selected individuals, comprising 12 physiotherapists, 4 occupational therapists, 11 caregivers, 9 children and 2 adolescents. Participants completed sociodemographic and clinical questionnaires and took part in semi-structured online interviews, focusing on two themes: Positive health approaches and the MAF tool. The study was approved by the local ethics committee (opinion 4.779.175).

### Results

Reflexive Thematic Analysis of the interviews resulted in two themes: (1) Perceptions regarding the MAF tool as an educational and contributory process to enhance the inclusion and participation of children and adolescents with disabilities, and (2) Barriers and facilitators for the implementation process of the MAF tool. The implementation of MAF was identified as a driving factor in promoting equity and increased participation of children and adolescents with disabilities in various settings, including health, education, and leisure.

**Data Availability Statement:** All relevant data are within the manuscript and its Supporting Information files.

**Funding:** This study was financed in part by the Coordenação de Aperfeiçoamento de Pessoal de Nível Superior - Brasil (CAPES) - Finance Code 001.

**Competing interests:** The authors have declared that no competing interests exist.

Interviewees highlighted the need to confront attitudinal, communication, and social barriers that may hinder the implementation of the tool.

## Conclusion

The implementation of the MAF tool was perceived as an innovation due to its focus on the abilities of individuals with disabilities. However, there is a need to restructure it to broaden its scope and access to different contexts in order to confront barriers and enhance the inclusion and participation of children and adolescents with disabilities.

## Introduction

In 2020, as the coronavirus (COVID-19) pandemic spread across the world, public health strategies, including social distancing measures, were implemented, causing disruptions in people's routines [1–5]. For children and adolescents with disabilities and their families, such measures interfered with access to healthcare, which is essentially provided through rehabilitation services. These measures brought visibility to and exacerbated existing difficulties, sparking discussions and demanding innovative actions, inclusive approaches, and those grounded in ensuring the rights of persons with disabilities [2,6,7].

Given this context, Schiariti and McWilliam (2021) [2] suggested measures for attention to persons with disabilities that can be implemented virtually, using low-cost online platforms suitable for children: (1) establishing collaborative and meaningful goals, (2) focusing on abilities, (3) empowering families for healthcare, (4) emphasizing the right of children and adolescents to express their feelings and opinions, (5) promoting emotional connections between professionals and families, (6) changing attitudes towards disability, (7) improving the provision of remote services, and (8) applying early intervention beyond the age of five [2].

In this regard, the educational tool My Abilities First (MAF), conceived by Verònica Schiariti and disseminated in 2020, aims to achieve these measures of attention to people with disabilities. MAF is grounded in the use of positive language and the process of informing professionals, family members, and the general public about the importance of positive attitudes towards people with disabilities [6]. According to Schiariti (2020a), the use of positive language, associated with the MAF approach, means paying attention to what the person with a disability can do. MAF proposes the adoption of an abilities identification card, the My Abilities ID CARD [2,6]. Furthermore, it is a proposal that, by valuing the abilities of children and adolescents with disabilities, can contribute to increased community engagement and effective participation [6,8–10].

Historically, approaches directed towards people with disabilities have neglected functional aspects, contextual influences, and the importance of family in the process [6,7]. These biases become more evident when one observes that the functional abilities of individuals with disabilities are often assessed only by professionals. Additionally, there is frequently limited collaboration between professionals and families, hindering the attainment of a more comprehensive and accurate view of the abilities of children and adolescents with disabilities in a wider range of situations and environments [8–12].

Furthermore, family-centered interventions are globally recommended, yet the focus on the individual and the clinic still prevails [6,9]. In recent years, studies indicate an increase in families seeking to express their needs for involvement in healthcare decision-making. Studies highlight this importance, as well as involving these families in information sharing [12–16]. In a study by Bamm and Rosenbaum (2008) [17], families considered access to information as

paramount, while professionals deemed providing education essential to better promote family-centered care, thus offering better quality services for the health of children and youth with disabilities [17].

Therefore, considering the guidelines of the Convention on the Rights of Persons with Disabilities in addressing these issues, it becomes evident that promoting positive health actions plays a fundamental role in stimulating innovations and empowering people with disabilities [18,19]. This emphasizes the importance of ensuring that individuals with disabilities receive attention widely, equally, and equitably compared to the rest of the population [18–21].

Given the above, since 2021, our research group has initiated a collaboration to implement the MAF tool for the first time in Brazil. Therefore, the study's objective was to analyze the participants' perceptions regarding the implementation of the MAF tool in Child Rehabilitation Centers (CERs). We believe that exposing the participants' experiences and interpretations regarding these topics can generate reflections and future actions that surpass our initial purposes. It goes beyond the attention on physical structure and body function but focuses on achieving better levels of activity and participation among children and adolescents with disabilities in different contexts.

## Materials and method

### Study design

This study follows a qualitative, descriptive, reflexive, and interpretative approach. The choice of Reflexive Thematic Analysis (RTA) aligns with the qualitative principles of this study, particularly considering the need to analyze responses from semi-structured interviews. Additionally, RTA was selected for emphasizing the importance of active participation and the researchers' theoretical grounding by immersing themselves in the data to address the research questions [22,23].

### Study setup

The data was collected from three distinct groups of participants (professionals, caregivers, and children/adolescents) who attended Specialized Child Rehabilitation Centers (CERs) located in three municipalities in the state of Rio Grande do Norte (RN), in the northeastern region of Brazil: Macaíba, Parnamirim, and Natal.

### Ethical aspects

The project was approved by the research ethics committee of the Faculty of Health Sciences of Trairi (Federal University of Rio Grande do Norte/Brazil—CAAE: 43972321.1.0000.5568 and opinion 4.779.175). Volunteers signed the consent form for images and audio authorization and the Informed Consent Form (ICF), following Resolution No. 466/2012 of the National Health Council/MS, which regulates research involving human subjects.

### Study phases and period

The study was organized into 5 main stages. After signing the ICF, participants completed a sociodemographic and clinical questionnaire. Subsequently, they took part in semi-structured interviews and workshops. These activities occurred from September 2022 to March 2023 (Fig 1).

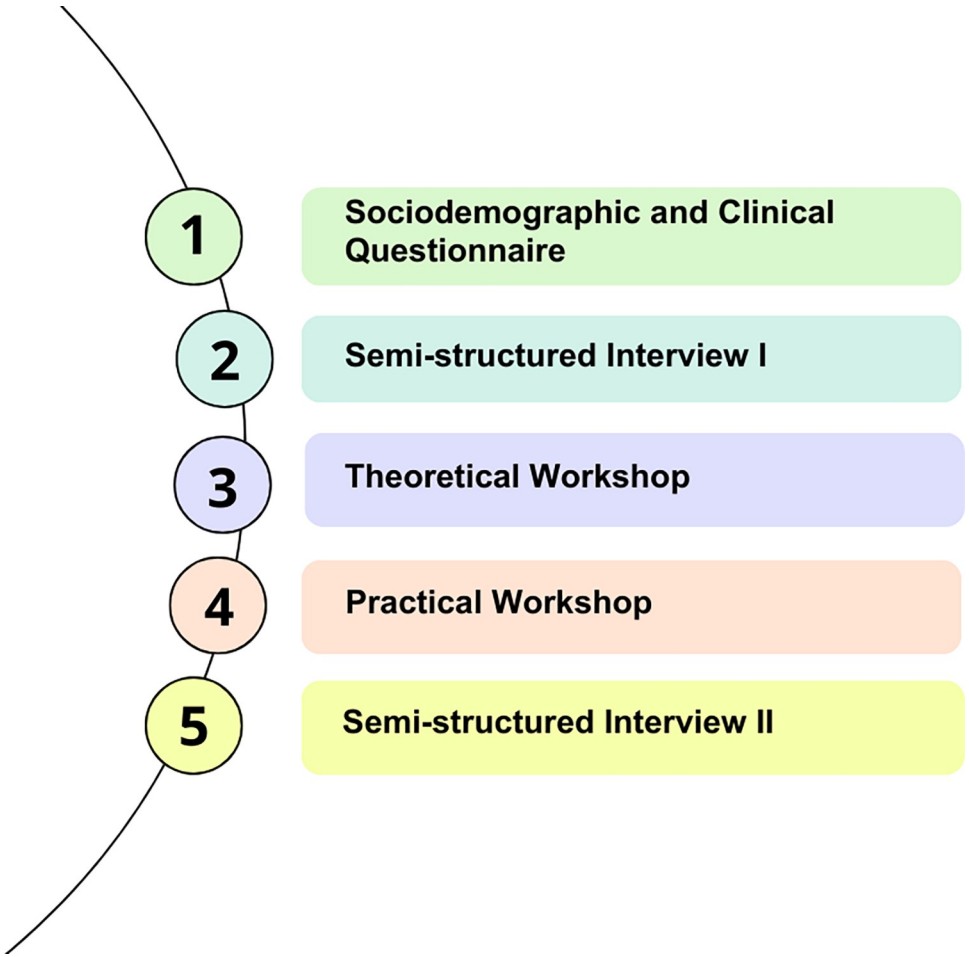

**Fig 1. Study stages.** Source: Research Data.

### Participants and eligibility criteria

Participant recruitment was intentionally conducted [24–26]. Selection took +place through active search in the Child Rehabilitation Centers (CERs) by the research team members, utilizing direct personal contact, phone calls, and messaging apps. The mentioned rehabilitation centers accommodated approximately 90 individuals with the potential to participate in the study. All of them were contacted; however, 27 participants met the inclusion criteria and responded to the semi-structured questionnaires and interviews I and II, after signing consent forms and authorization for images and audios.

The following inclusion criteria were considered for participants: (1) be a physiotherapist or occupational therapist involved in pediatric healthcare; (2) be a child or adolescent with disabilities, aged between 6 and 14, of all genders, and accompanied by a caregiver with some degree of kinship; (3) have an active registration in one of the selected CERs for the study (for professionals, children/adolescents); (4) understand and accepting the study proposal; and (5) be capable of accessing basic communication resources through electronic means to participate in planning, workshops, and to respond to online data collection instruments.

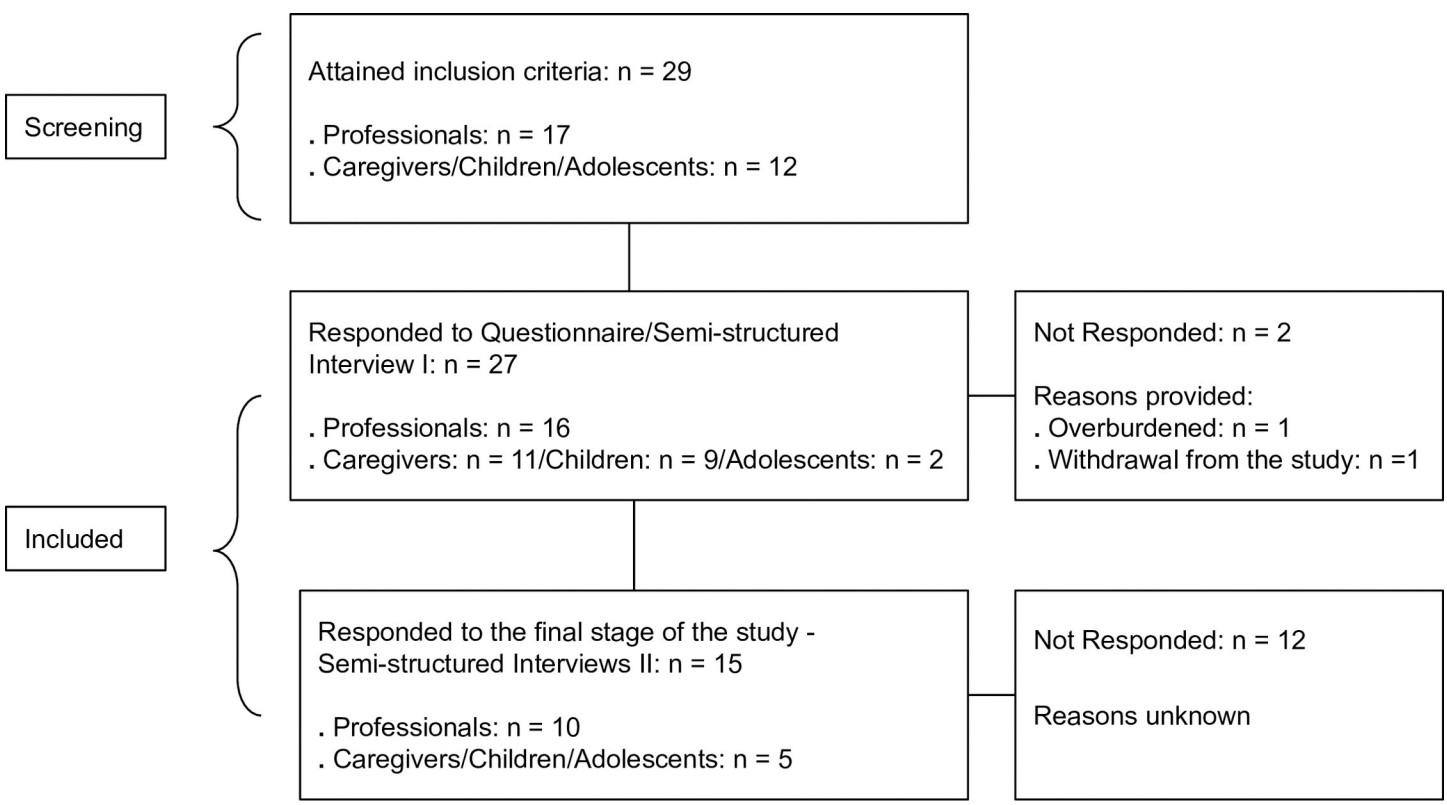

**Fig 2. Participants and eligibility criteria.** Source: Research Data.

Exclusion criteria included: (1) failure to respond to the sociodemographic questionnaire or semi-structured Interview I; (2) disengagement from the Child Rehabilitation Center (CER) to which the participant would be enrolled during the study period.

Twenty-seven participants were included and recruited to respond to semi-structured Interviews I and II. However, only 15 participants responded to semi-structured Interview II. Further details can be found in Fig 2.

### Data collection instruments

For data collection, the following were used: (1) Sociodemographic and clinical questionnaires and (2) Semi-structured interviews (I and II).

**Sociodemographic and clinical questionnaire.** The researchers developed questionnaires based on the study's themes and existing literature [6,27,28]. The questionnaire administered to professionals aimed to gather information on: (1) personal identification, (2) professional category, and (3) professional experience history, clientele served, and interventions used with children/adolescents with disabilities (https://forms.gle/NSRfVxjRRcdUZ9nT6). The questionnaire directed towards children/adolescents and their caregivers aimed to collect information on: (1) personal identification, (2) socioeconomic and educational conditions, and (3) understanding of the child/adolescent's overall abilities (https://forms.gle/9axVTXcCBUhxhqWA7).

**Semi-structured Interview I and II.** Semi-structured Interviews I and II were administered to all participant groups. In semi-structured Interview I, conducted with caregivers and children/adolescents with disabilities, the objective was to explore the dialectical relationship

between positive health attitudes and topics related to valuing the abilities of children and adolescents with disabilities (https://forms.gle/qN2k9pZc5JMSfAfv8). However, in semi-structured Interview I applied to professionals, apart from this objective, the aim was to understand the feasibility of implementing MAF (https://forms.gle/iieRJEKNmVJQfjia8).

During semi-structured Interview II, information related to the perceptions of professionals (https://forms.gle/PU34GdRSmJ5gb8xW6) and families (https://forms.gle/oL7BBqCSeUebE24J6) about the implementation process of the MAF tool was sought. At this stage, open-ended questions were formulated by the researchers based on the study's themes [2,6,7].

Alphanumeric codes were assigned to the statements obtained from semi-structured Interview II to preserve the interviewees' identities. 'C' was used to represent family members (Caregivers, Children, and Adolescents), 'F' for physiotherapists, and 'OT' for occupational therapists, followed by numbers indicating the interview sequence in the system.

Time taken to respond to semi-structured Interview I ranged from 10 to 20 minutes, whereas for Interview II, it ranged from 20 to 40 minutes. All interviews were digitally recorded for literal transcription and data analysis.

## My Abilities First Tool

The MAF tool incorporates guiding principles for services aimed at children and adolescents with disabilities and their families. These principles encompass the valuing of capacity and abilities and the potential to achieve goals. Additionally, these involve the interactions between the individual and the environment, promote family-centered care, and adopt a biopsychosocial rights-based approach for assessments, planning, and interventions. Ultimately, they aim to empower both families and individuals with disabilities to participate in decisions related to their own care [2,6].

The MAF tool offers animations (videos) with objectives centered on people with disabilities: (1) The first video introduces the application of an abilities-oriented approach in healthcare encounters. It proposes the creation of an abilities identification card, the My Abilities ID Card. Target audience: healthcare professionals and related fields (https://youtu.be/WyW6ey3kHvM); (2) The second video highlights the importance of applying a holistic approach in routine health consultations through questions to identify abilities. Target audience: healthcare professionals, researchers, educators, and students (https://youtu.be/Dnn_-0IEe_Q); and (3) The third video promotes a change in attitude towards disability and individuals with disabilities. In this animation, a typically developing child advocates for the inclusion of a child with a disability. The target audience is the general public, primarily school-aged children with disabilities and their peers (https://youtu.be/myHFKggNeGc) [2,6].

## Educational interventions

Two workshops were conducted to share knowledge and facilitate active participation in the study, particularly in Semi-Structured Interview II.

The theoretical workshop aimed to present, discuss, and provide theoretical content on the My Abilities First theme and its implementation process in CERs. Scientific articles related to the study's theme and animations inherent to the MAF tool (mentioned in My Abilities First Tool section) were utilized and shared among researchers and participants.

The practical workshop, on the other hand, provided practical content with the following objectives: to create the Abilities Identification Card, the My Abilities ID Card, and to adopt strategies for using this card and disseminating the animations (professionals: during

**Table 1. Workshop content.**

| Workshop | Characterization of the activity | Content/Methodology | Venue for execution | Duration in minutes—groups of 6 to 8 partners/participants |
|---|---|---|---|---|
| Meeting I | Theoretical workshop (understanding and discussion of concepts, principles, models, and frameworks of the MAF tool) | Content: (1) Presentation dynamics of partners/participants and researchers. (2) Presentation of the My Abilities First (MAF) tool. (3) Presentation of guidelines for implementing the MAF tool. Methodology: (1) Conversational circles, Video lessons and slides. (2) Use of didactic resources about MAF (available on the web, in articles). (3) Providing video lessons, links, articles on MAF tool content. | on—line | 40–60 |
| Meeting II | Practical workshop (Creation of the My Abilities ID CARD) | Content: (1) Creation of the My Abilities ID CARD. Methodology: (1) For the making of the My Abilities ID CARD: use of manual resources (printed materials, drawings, and writing) and technological resources (online design platforms and visual communication). | In person | 60–90 |

**Source:** Study data.

appointments; caregivers and children/adolescents: at home, school, within the community, and in healthcare services). For further information, please refer to Table 1.

## Data analysis

The quantifiable data from the sociodemographic/clinical questionnaires and Semi-Structured Interview I were analyzed and presented by calculating the percentages in relation to the total observations in the dataset. Non-quantifiable data (verbal responses) were transcribed in full for presentation in figures. Further information can be found in the results section of this study.

The analysis of semi-structured interviews II was based on Reflective AT and was conducted by the main researcher (RFA) and coordinating researcher (ARRL), assisted by an experienced collaborator (CCAFJ) [23,24]. For better interpretation of the data and preparation of the report, the process was outlined according to the six stages of thematic analysis by Braun and Clarke (2006;2013;2019) [22,25,26], presented in Table 2.

The entire process, including the six stages (as in Table 2), will be described below and is documented and archived for clarity and traceability, consequently ensuring greater rigor and reliability of the research [28–30].

Initially, in the familiarization stage, the interviewees' speeches were transcribed in full by three volunteers (TFCP, SSVM and MCLC), checked by the researchers (RFA, ARRL and CCAFJ), and the interview participants themselves (Table 2). Subsequently, the transcribed speeches were organized into a Word document. Readings and rereadings of all the material continued throughout the entire process, but primarily at this stage.

In the subsequent stages, the coding process and the construction of themes predominantly followed an inductive approach, aiming to remain faithful to the participants' responses. Nevertheless, a certain degree of deductive analysis was employed, emphasizing the importance of recording the subjectivity of less apparent factors in the responses [28–30].

Table 2. Stages of thematic analysis.

| Stage | Description | Who carried out |
|---|---|---|
| 1. Familiarization with data | Transcription and review; reading and rereading the file; and registering ideas. | Interview Transcriptions: TFCP, SSVM and MCLC Transcription Supervision: RFA Transcription Conference: Interviewed participants of the study, RFA, ARRL and CCAFJ Reading, Rereading, and Idea Logging: TFCP, SSVM, and MCLC, RFA, ARRL and CCAFJ |
| 2. Generating initial codes | Code relevant aspects and gather important extracts from each code. | RFA and CCAFJ |
| 3. Searching for themes | Group codes in potential themes and correlate data according to each highlighted theme. | RFA and CCAFJ |
| 4. Reviewing of themes | Verify the correlation of themes and whether they influence the extracts and database; structure a thematic map of the complete analysis. | RFA, ARRL and CCAFJ |
| 5. Definition and naming of themes | Structure and restructure each theme and history of the analysis; designate and register each theme concisely. | RFA, ARRL and EL |
| 6. Producing the report | Operationalize and register the real examples experienced; analyze all selected extracts related to research questions and literature; and structure the scientific report of the study. | RFA and ARRL |

Source: Adapted from Braun and Clarke (2006;2013;2019) [22,25,26] for structure and definition of study stages.

The themes were identified through predominantly semantic coding, meaning that the description of the speeches did not require continuous exploration of hidden meanings, due to the clarity of these speeches and the focus on a specific theme, which was the implementation process of the MAF tool. However, there was some latent level coding when it was necessary to employ a more interpretative exercise of the speeches, considering, to justify this need, the participants' premises, context, and social position, such as speeches with the association of regional terms in unfinished sentences.

The analysis was dynamic and required, following Braun and Clarke (2020) [24], observations, records, revisions, exclusions, and new inclusions of content, that is, it required moving forward and/or returning [24]. However, the basis of the process was sequenced throughout the thematic analysis [23–25].

For the recording and construction of each theme, the process involved the study's literary foundation, the expertise of the researchers and collaborators (RFA, ARRL and EL) to generate codes, gathering relevant excerpts, grouping them into potential themes and sub-themes, and thus reaching the stage of definitively naming the themes (Fig 3). With this refinement, in line with Braun and Clarke (2019) [22], to achieve the final map of the data analysis process, 11 codes encompassed the breadth of ideas and supported the definition of themes and the report writing, contributing to answering the research question and thus meeting the overall study objective. As a result, two themes emerged: Theme 1 - 'Perceptions of the MAF tool as an educational and contributory process to enhance the inclusion and participation of children and adolescents with disabilities,' and Theme 2 - 'Barriers and facilitators for the implementation process of MAF' (Fig 3).

## Results

The study involved 12 physiotherapists, 4 occupational therapists, 11 caregivers, 7 children, and 2 adolescents. The majority were professionals with 11 or more years of experience in

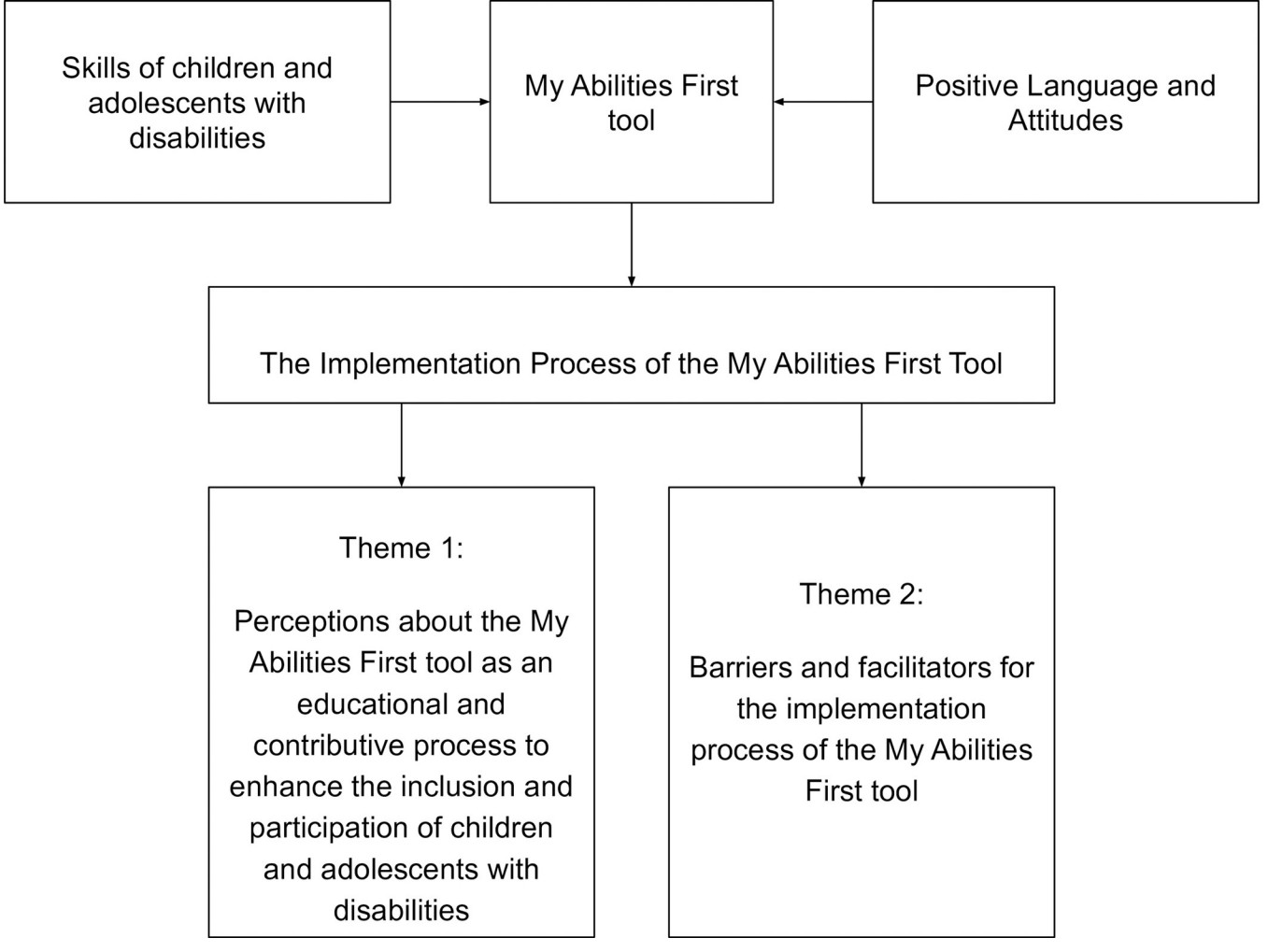

**Fig 3. Data analysis process map.** Source: Research data.

pediatric care (66.7%), physiotherapists (73.3%), and within the age range of 30 to 49 years old (81.4%). Only 33.3% reported participating in activities related to positive language in health, a foundational aspect for the MAF tool. Further details are presented in Fig 4.

A total of 11 children/adolescents with their caregivers participated in the study. Diverse sociodemographic and clinical characteristics were recorded, and percentages were calculated based on the numerical values assigned to each question, aiming for better exposure and transparency of the obtained data. The predominant age of the children and adolescents was 7 years (54.5%), and all attended regular education. Further sociodemographic information of this group is available in Table 3.

In this questionnaire, children, adolescents, and their caregivers answered questions about their abilities. There was a greater appreciation of the abilities of children and adolescents within the home (50% always and 20% often) compared to outside it (20% always and 10% often). When asked about what should happen to value these abilities and about the abilities they would like to develop, there was a variety of responses based on the specific needs of these participants. The responses are presented in Fig 5.

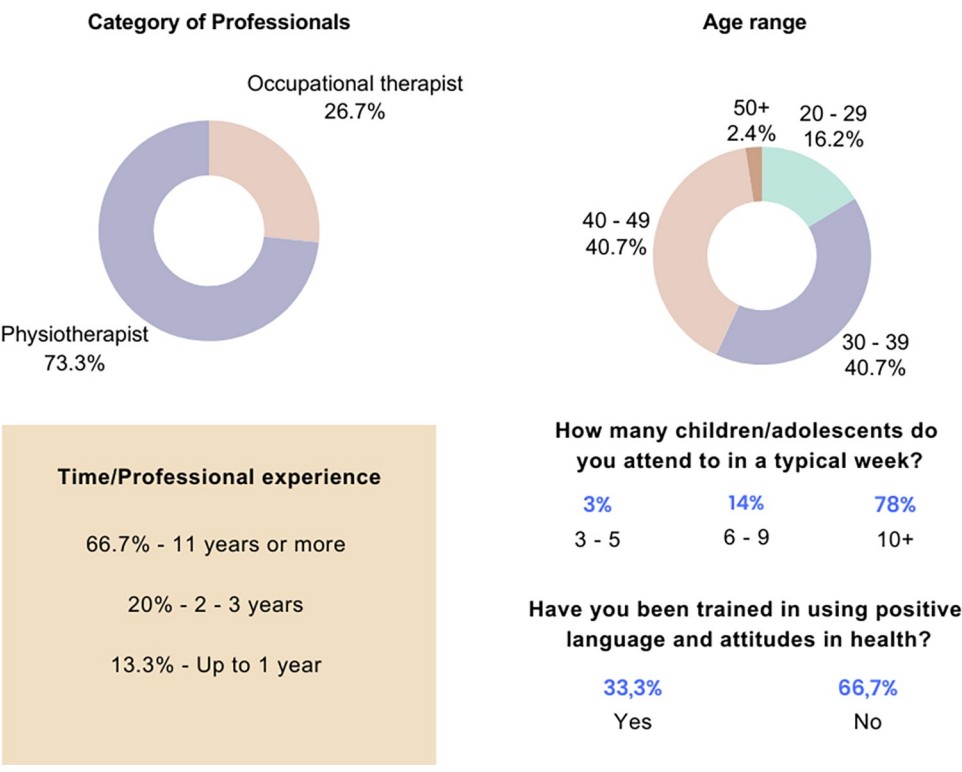

**Fig 4. Data from the sociodemographic and clinical questionnaire applied to professionals.** Source: Research data.

Regarding the responses from semi-structured interview I, it was identified that although 60% of professionals claimed to be familiar with positive language and attitudes in health and considered it feasible to apply in their daily practices, just over half of them (36.4% always and 18.2% often) address the abilities of interest of the child/adolescent in their treatment programs. Further information is presented in Fig 6.

The children/adolescents and caregivers were unanimous in emphasizing the importance of valuing abilities. For 90% of the families, professionals encourage children/adolescents with disabilities to acknowledge their own abilities, and more than 50% of the families report that professionals value the abilities of interest of the children/adolescents (Fig 7).

The analysis of Semi-structured Interview II was based on Reflexive Thematic Analysis and resulted in the construction of two themes grounded in 15 interviews (10 professionals and 5 families), namely: (1) Perceptions about the MAF tool as an educational and contributory process to enhance the inclusion and participation of children and adolescents with disabilities, and (2) Barriers and facilitators for the implementation process of MAF.

The first theme was constructed from responses regarding the knowledge of the MAF tool and its usefulness. Interviewees demonstrated an understanding of the tool's proposal and considered it applicable. Concerning professionals, the majority agreed that the MAF tool could be useful in healthcare for children and adolescents, highlighting the positive impact of the possibility to change the dynamics between interventions that focus on disability and those that value abilities. Furthermore, they pointed out the MAF tool as a means to promote changes in professional and family behaviors regarding the perception of abilities.

"Yes. [. . .] To stop focusing only on limitations, illness, or disability and see the potentialities [. . .]. Specifically regarding physiotherapy, this tool can serve as a guide [. . .]. Besides, I

Table 3. Characteristics of participating children and adolescents in the study.

| Demographic and clinical profile of the children/adolescents with disabilities included in the study | n | % |
|---|---|---|
| *Characteristics of children/adolescents with disabilities.* | | |
| Ages: | | |
| 7 years old | 6 | 54,5 |
| 8 years old | 1 | 9,1 |
| 10 years old | 1 | 9,1 |
| 11 years old | 1 | 9,1 |
| 14 years old | 2 | 18,2 |
| Gender: | | |
| Female | 7 | 63,6 |
| Male | 4 | 36,4 |
| Currently studying: | | |
| Yes | 11 | 100 |
| Education level: | | |
| Early childhood education | 4 | 36,4 |
| Elementary school | 7 | 63,6 |
| Classroom assistant: | | |
| Yes | 7 | 63,6 |
| No | 4 | 36,4 |
| Health condition: | | |
| Spina bifida | 1 | 9,1 |
| Autism | 1 | 9,1 |
| Brachial Plexus Palsy (BPP) | 2 | 18,2 |
| Cerebral palsy | 7 | 63,6 |
| Locomotion: | | |
| Walks without using devices | 5 | 45,5 |
| Walks with devices | 1 | 9,1 |
| Walks with supervision and/or support from others | 1 | 9,1 |
| Wheelchair | 4 | 36,4 |
| Communication: | | |
| Fluent speech | 6 | 54,5 |
| Speech or communication with difficulty/needs assistance | 5 | 45,5 |
| Object manipulation: | | |
| Independent | 10 | 90,9 |
| Needs assistance | 1 | 9,1 |

Source: Research data.

think it can also help families to have another vision about disability [. . .] families cling to unrealistic expectations [. . .] It is necessary to respect the child as a being who has desires [. . .] not to deprive. . . and not necessarily meet the expectations of parents or society. . . in a certain pattern." (F1)

"Yes, I think it's very useful to change the perspective of therapists and families and consequently of children and adolescents about disability and the individual." (TO2)

However, the participants expressed concerns regarding the credibility of the feasibility of the proposal, which requires behavioral and cultural changes that often depend on public

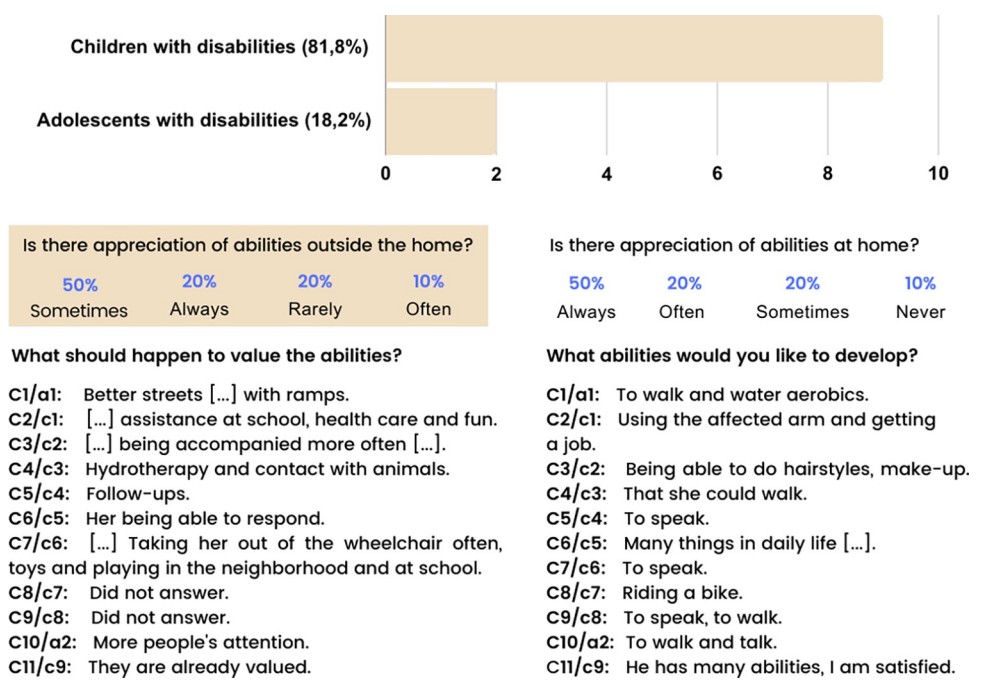

**Fig 5. Data obtained from the sociodemographic and clinical questionnaire responses from Children/Adolescents (c/a) and their Caregivers (C).** Source: Research data.

policies. This uncertainty about the implementation and usefulness of the tool was evident in the statements of the families, such as C1 and C2.

"I think so. But, there needs to be a way to use it." (C1)

"Yes, but it needs to go beyond just being theoretical." (C2)

The participation of the child or adolescent with the caregiver in the same space with the professionals was considered important for the implementation process of the MAF tool, according to these families, regardless of the category (families or professionals). They further

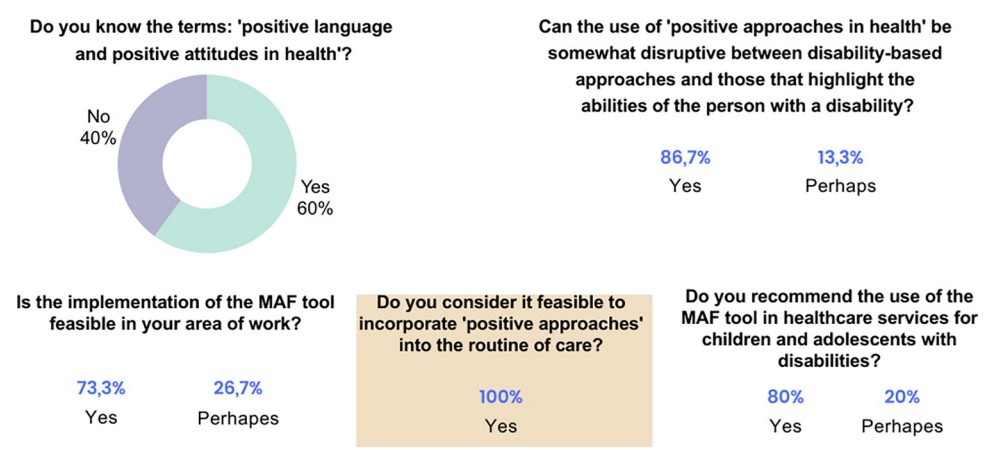

**Fig 6. Data from semi-structured Interview I applied to professionals.** Source: Research data.

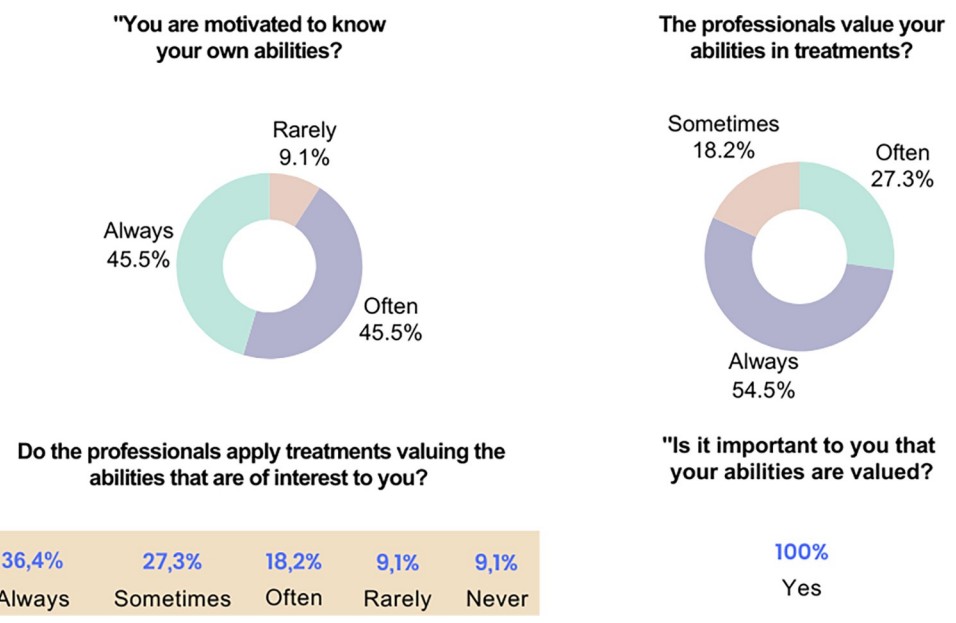

**Fig 7. Data from semi-structured Interview I applied to children/adolescents and caregivers.** Source: Research data.

added that the tool itself provides this educational and inclusive opportunity, involving the acquisition and sharing of knowledge.

> "Yes, it's useful, very good to be able to talk about the abilities that a child with a disability has, and to be able to do it in the way that they know." (C3)

> "Yes, and because we are being listened to more, it's good to know that my opinion matters."(C4)

In general, participants highlighted positive aspects related to the MAF tool. However, concerns about the potential for increased discrimination, ableism, among others, were expressed in statements throughout the study, particularly in the statement of C5 below, which illustrates this context and has the potential to stimulate further reflections:

> "No! I think it's unnecessary to invest in the idea of highlighting the abilities of the person with a disability in the first place, because it can further highlight the disability."(C5)

Interviewees showed, at times, a certain disinterest and difficulty in expressing opinions due to insufficient knowledge about the tool. However, when analyzing strategies to improve the tool, professionals emphasized the pressing need for an evidence-based educational process and behavioral changes.

> "Yes. Once the potentialities are visible, it diminishes the preconceived notions that disabilities only accompany difficulties. It opens up the perception that there are also facilities, and through them, responses, interaction, and understanding can be established to set goals in caring for children and adolescents with disabilities." (F2)

> "Yes. From knowing the abilities of children with disabilities, we, as therapists, can work to further enhance such abilities." (F4)

"Yes, as by initiating the treatment and enhancing the abilities, the patient x therapist bond is established in less time, the family learns to value the present abilities from the beginning and how to stimulate them, and finally, we put the patient on stage and active from the start of the sessions." (TO3)

For the interviewees, the enhancement of the MAF implementation process requires dissemination, use in various contexts, and for a wide audience. From the perspective of these interviewees, restricting access to the tool also means limiting access to knowledge and the processes of inclusion of people with disabilities.

Using it everywhere and not just in healthcare settings." (C1)

"Explaining it to people everywhere." (C2)

"Taking it to schools, public transportation stops, neighborhoods." (C3)

"It works if used for those with disabilities and also for those without disabilities." (C5)

The school was mentioned as the appropriate place for the dissemination and use of the tool due to the daily interaction with people outside the family circle. Among professionals and families, similar responses were observed regarding what is necessary for the implementation process of MAF. Access to information, extensive dissemination, and usage in different contexts were emphasized.

"I believe that thinking of ways to make it something that encompasses everyone and not just people with disabilities could be positive. It would be a way to put everyone's abilities into perspective, without necessarily being a tool used only by people with disabilities." (F2)

"By spreading it to the largest number of professionals in the field, families, and including it in assessment protocols and anamnesis." (TO2)

Regarding the responses, which formed the basis for the construction of the second theme: 'Barriers and facilitators for the implementation process of MAF,' it was observed what is possible for the mentioned process to succeed.

Families were assertive in highlighting the need for resolution-oriented actions for usage. However, participants emphasized the need to avoid using the MAF tool to highlight individuals with disabilities and thereby generate discrimination and other stigmas. Additionally, participants suggested the importance of ensuring training for the proper use of the tool.

"Using it to highlight the child, discriminate." (C1)

"Staying on paper." (C2)

"Using more rigor, more 'authority' to make the patient trust, to make the patient feel secure." (C5)

"[. . .] if the therapist is not confident about the tool, there may be some bias." (F7)

When asked about what could go wrong in using the MAF tool, participants highlighted the need to address barriers related to disability and the life of a person with disabilities, such as attitudinal, communication/information, political, and social barriers.

"The stigma of being." (TO1)

"I don't see how it could go wrong if well indicated." (F1)

"I am concerned that the activities listed in My Abilities First might be perceived as what the person can do and that this ends up 'defining' who they are and what they can or cannot do. It needs to be clear that these are ways to interact and demonstrate what one likes, what one does, but that it does not limit the person to what they can do." (F2)

"In my opinion, nothing can go wrong, as long as it is applied with common sense, caution, and according to reality." (TO2)

"Difficulty for children to express what they like to do." (F3)

"The family is not adhering." (F5)

When asked to identify reasons to recommend the adoption of the MAF tool, families shared arguments related to the purposes of the tool:

"s will be recognized." (C1)

"See abilities before the disability." (C2)

"One reason to recommend is that yes, it is possible for a child with a disability to do it their own way." (C3)

"I recommend it because it helps people with disabilities to give their opinion on how it can improve." (C4)

"If it gives more security in the application, it will generate more results." (C5)

The reasons for recommending the tool, pointed out by the professionals, were accompanied by the desire to restructure the tool and expand objectives. However, the recommendations were to facilitate more humane interventions and promote equity, as well as enhance inclusion.

"Facilitation" (TO1)

"The card with the abilities could be used when the child has difficulty communicating, which would help include them in activities at school, for example, respecting what they like and are able to do." (F1)

"Facilitating the understanding that the universe of a person with disabilities is not just limited, as the tool 'my abilities first' proposes, can assist these individuals in various areas of their lives, from meeting a new therapist to the school environment, through family, community, and the entire surroundings. Getting to know the child better through their abilities makes interaction and social inclusion easier." (F2)

"Self-esteem, seeing individuals with disabilities as individuals of possibilities." (TO2)

"Shifting the intervention focus from structure and function to activity and participation." (F3)

"Valuing children and young people with disabilities and further enhancing their abilities." (F4)

"Seeing the child as a person, not a pathology." (F5)

"In my view, the primary purpose is to put the patient in the spotlight right from the first interaction, active and participative, as families sometimes fail to notice certain present abilities or what they can do in their own way." (TO3)

The respondents' answers reflect a variety of perceptions about the implementation and usefulness of the MAF tool. While some highlighted potential benefits, such as increased self-esteem, appreciation of individual abilities, and contribution to promoting inclusion, others expressed concerns about possible adverse effects, such as increased discrimination and stigma. There was also a common emphasis on the importance of a person-centered approach, recognizing individual abilities and the importance of active participation of individuals with disabilities in their communities. These diverse perspectives offer valuable insights to guide future research and practices in the field of children and adolescents' rehabilitation.

## Discussion

This study, which describes the first experience of implementing the MAF tool in Brazil, aimed to analyze the perceptions of physiotherapists, occupational therapists, children, adolescents with disabilities, and their caregivers regarding the implementation of the My Abilities First (MAF) tool in Pediatric Specialized Rehabilitation Centers (CERs).

The MAF is seen as an innovative tool because it emphasizes the individual abilities of people with disabilities, rather than focusing on their disabilities, and it is aligned with the United Nations Conventions on the Rights of the Child and on the Rights of Persons with Disabilities, embracing the rights of self-determination, autonomy, and dignity of people with disabilities [2,6,18,19].

The heterogeneity of the participants (age range, cultural and social aspects) impacted the meticulous development of the data collection instrument questions and applicability. However, the study provided an integrative environment in which perceptions about the MAF tool and its implementation were shared.

Regarding Theme 1: 'Perceptions of the MAF tool as an educational process and contribution to enhance the inclusion and participation of children and adolescents with disabilities,' participants demonstrated understanding of the tool and considered it applicable in their contexts, whether in therapeutic planning, in the services where they work, or in their daily lives, as mentioned by Schiariti (2020) [6] when referring to the tool's accessibility in different aspects.

Both participant groups emphasized the importance of educational workshops and 'listening' to the children/adolescents and families in the study. These aspects allow for greater integration and participation not only of the child or adolescent but also of the family in the therapeutic context, as evidenced by Palisano et al. (2009) and Almasri et al. (2011) [14,15]. This approach facilitates the broad involvement of the family in the therapeutic process, significantly contributing to adherence to services, therapeutic planning, and consequently, the use of the proposed tool.

The inclusion of children/adolescents with disabilities and their families in the planning or any other stage of a therapeutic plan or even valuing their abilities, are positive aspects related to the MAF tool. These aspects favor the empowerment of this population and make them protagonists of their rehabilitation. These data are aligned with the discussions of Schiariti & McWilliam (2021), Schiariti (2020), and Palomo-Carrión et al. (2022) [2,6,10]. These authors, in mentioning collaborative and empowering approaches for children and adolescents with disabilities, emphasize the importance of focusing on abilities and respecting the rights of people with disabilities to express feelings, opinions, and their preferences.

Another point highlighted by the participants in their responses was the need to expand the scope of the MAF tool. The most suggested strategy was that of broad information, as they believed in promoting knowledge, as a means to valorize abilities, inclusion, and effective participation of people with disabilities [8,9,11]. This possibility was mentioned to reach a broad and diversified audience.

Still within the context of Theme 1, participants' perceptions highlight the importance of extending the reach of the MAF tool to other environments frequented by typical and atypical children, to better serve the purpose of increasing engagement, fostering more effective participation, and creating new and promising perspectives for children and adolescents with disabilities [20,21]. In this sense, its use in the school environment was identified as particularly notable. Regardless of the presence of disability, respondents indicated the use for everyone, aiming to promote better inclusion and integration of children and adolescents with disabilities with their peers.

Thus, we emphasize that the implementation of the MAF tool was referred to as desirable, useful, and recommended. However, it requires widespread dissemination, use in various contexts, and an expanded target audience.

In the scope of Theme 2, entitled 'Barriers and Facilitators for the Implementation Process of MAF,' the emphasis was on the importance of promoting behavioral changes and attitude changes. Participants pointed out barriers related to disability and the life of people with disabilities. These barriers were associated with the implementation of the MAF tool and the means to address them, among others, through health education.

The need to instigate governmental actions was expressed in the study. The allocation of government resources can be positive for applying the principles of equity, justice, and respect for human rights. Additionally, it can promote process efficiency, the exchange of knowledge and experiences, as well as ensure an integrated and holistic approach to promoting broad access to health and inclusion with effective participation of people with disabilities, corroborating with proposals by Schiariti and McWilliam (2021) [2] and Schiariti (2020) [6].

Therefore, professionals and families, by emphasizing the importance of health education, underscored the need to transform discussions about the implementation of MAF into concrete practices, beyond the theoretical scope and implementation in healthcare institutions. This argument was echoed in the statements of the interviewees, advocating for broadening perspectives and adopting positive health attitudes. Consequently, the aim is to maximize functional aspects and abilities while minimizing the constant focus on highlighting the disability. These considerations find support in the World Health Organization's (WHO) Convention on the Rights of Persons with Disabilities, which highlights that positive attitudes in the field of health play a fundamental role in encouraging innovation and empowering people with disabilities [18].

The implementation of the MAF tool may face challenges, with resistance to attitude changes being one of the most significant, especially when prioritizing the recognition of the abilities of people with disabilities over a disability-focused approach. This process requires a deeper understanding of the MAF tool and the development of strategies for the effective use of the My Abilities ID CARD. To overcome these challenges, actions such as continuous education for professionals and health education for healthcare service users, as well as for the general population, play crucial roles. These measures not only support the adoption of the most appropriate strategies for the implementation process but also promote a broader and more inclusive understanding of the abilities of people with disabilities.

However, the study revealed new insights at each stage. This brought additional perspectives from the interviewees to light, such as the implementation of MAF in various healthcare centers and schools, to meet the need for an effective transition from disability-focused

approaches to approaches that promote the rights of people with disabilities, which is in line with Schiariti and McWilliam (2021), Schiariti (2020), and Liao et al. (2023) [2,6,10]. Additionally, when interviewees mention the importance of policy-related actions, they hope to ensure that children/adolescents with disabilities can showcase their abilities, regardless of the nature of the disability, and without the constant need to prove relevant or specific degrees of incapacity to secure their rights.

## Final considerations

The discussions centered on the perceptions of a heterogeneous group of participants regarding the implementation process of the My Abilities First (MAF) tool were conducive to new analyses, thereby enriching the research journey. In this context, observations arose that had not previously been considered by the researchers. These observations are related to the need to expand the use and acceptance of the MAF proposal through restructuring the tool for use in different contexts and through health education actions.

The innovative, accessible, and low-cost approach proposed by the tool focuses on valuing the abilities of people with disabilities and proves to be an achievable and promising process for improving clinical practice in healthcare, changing perceptions, behaviors, and contributing to addressing the barriers associated with disability and the lives of people with disabilities. In summary, this formed the basis for the study, which also provided new perspectives and interpretations on the implementation process of the MAF tool, such as expanding implementation beyond the healthcare context.

Based on these results and considering the limitations outlined in the study, the lack of greater diversity in the sample in terms of cultural, socioeconomic, and primarily geographical differences, as well as the absence of cross-cultural adaptation of the MAF tool, may have affected obtaining a better perspective on the results. These two aspects may be crucial for promising directions in future research.

Furthermore, other important perspectives to be included in future studies include the formulation of strategies for the use of the My Abilities ID CARD and the development of protocols or implementation manuals for the MAF.

## Supporting information

**S1 File.**
(XLSX)

## Acknowledgments

We would like to thank the Brazilian Coordination for the Improvement of Higher Education Personnel, the Secretariat Public Health of Rio Grande do Norte and the Postgraduate Program in Physiotherapy at UFRN.

## Author Contributions

**Conceptualization:** Roselene F. Alencar, Egmar Longo, Verónica Schiariti, Monique L. G. Coelho, Ana Raquel R. Lindquist.

**Data curation:** Roselene F. Alencar, Egmar Longo, Verónica Schiariti, Caline C. A. F. Jesus, Rocío P. Carrión, Carolina D. L. Alvarez, Monique L. G. Coelho, Tatiana F. C. Pereira, Maria C. L. Cruz, Sanzia S. V. Melgão, Ana Raquel R. Lindquist.

**Formal analysis:** Roselene F. Alencar, Egmar Longo, Verónica Schiariti, Caline C. A. F. Jesus, Carolina D. L. Alvarez, Tatiana F. C. Pereira, Maria C. L. Cruz, Ana Raquel R. Lindquist.

**Investigation:** Roselene F. Alencar, Ana Raquel R. Lindquist.

**Methodology:** Roselene F. Alencar, Egmar Longo, Verónica Schiariti, Caline C. A. F. Jesus, Rocío P. Carrión, Carolina D. L. Alvarez, Monique L. G. Coelho, Ana Raquel R. Lindquist.

**Project administration:** Roselene F. Alencar, Ana Raquel R. Lindquist.

**Supervision:** Ana Raquel R. Lindquist.

**Visualization:** Egmar Longo, Verónica Schiariti, Caline C. A. F. Jesus, Rocío P. Carrión, Carolina D. L. Alvarez, Monique L. G. Coelho, Tatiana F. C. Pereira, Maria C. L. Cruz, Sanzia S. V. Melgão, Ana Raquel R. Lindquist.

**Writing – original draft:** Roselene F. Alencar, Egmar Longo, Verónica Schiariti, Caline C. A. F. Jesus, Rocío P. Carrión, Carolina D. L. Alvarez, Ana Raquel R. Lindquist.

**Writing – review & editing:** Roselene F. Alencar, Egmar Longo, Verónica Schiariti, Caline C. A. F. Jesus, Rocío P. Carrión, Carolina D. L. Alvarez, Monique L. G. Coelho, Tatiana F. C. Pereira, Maria C. L. Cruz, Sanzia S. V. Melgão, Ana Raquel R. Lindquist.

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
