## [Decision Letter · Decision Letter 0]

12 Jan 2024

PONE-D-23-41828Implementation of the My Abilities First Tool: a qualitative study on the perceptions of professionals, caregivers, children, and adolescents with disabilitiesPLOS ONE

Dear Dr. Alencar,

Thank you for submitting your manuscript to PLOS ONE. After careful consideration, we feel that it has merit but does not fully meet PLOS ONE’s publication criteria as it currently stands. Therefore, we invite you to submit a revised version of the manuscript that addresses the points raised during the review process.

We look forward to receiving your revised manuscript.

Kind regards,

Renato S. Melo, PhD

Academic Editor

PLOS ONE

Journal Requirements:

2. We note that your Data Availability Statement is currently as follows: “All relevant data are within the manuscript and its Supporting Information files.”

Additional Editor Comments (if provided):

Dear authors, the reviewers have finished reviewing the manuscript and decided that it has merit to be accepted for publication, however, it needs some adjustments before that.

Reviewers' comments:

Reviewer's Responses to Questions

**Comments to the Author**

1. Is the manuscript technically sound, and do the data support the conclusions?

Reviewer #1: Yes

Reviewer #2: Yes

Reviewer #3: Yes

2. Has the statistical analysis been performed appropriately and rigorously? 

Reviewer #1: Yes

Reviewer #2: Yes

Reviewer #3: Yes

3. Have the authors made all data underlying the findings in their manuscript fully available?

Reviewer #1: Yes

Reviewer #2: Yes

Reviewer #3: Yes

4. Is the manuscript presented in an intelligible fashion and written in standard English?

Reviewer #1: Yes

Reviewer #2: Yes

Reviewer #3: Yes

5. Review Comments to the Author

Reviewer #1: The manuscript presents a comprehensive study on the perceptions of professionals, caregivers, children, and adolescents with disabilities regarding the implementation of the My Abilities First (MAF) tool in Specialized Child Rehabilitation Centers (CERs). Overall, the manuscript makes a valuable contribution to the field of rehabilitation by shedding light on the perceptions of various stakeholders regarding the MAF tool. I have some suggestions and comments designed to improve this manuscript:

Introduction:

-The introduction briefly mentions the MAF tool and its grounding in positive language, valuing the potential of children and adolescents with disabilities. Adding a concise description or definition of the MAF tool would enhance understanding and provide clarity on its intended impact.

Method:

- While the educational interventions are briefly mentioned, providing more details on the content, structure, and objectives of the theoretical and practical workshops would enhance the reader's understanding of their role in the study.

-The authors need to further clarify code identification to illustrate the semantic and latent ways of coding to assist in understanding the analytical process more concretely.

-Given the emphasis on reflexive analysis, it would be valuable if the authors include a brief discussion on the researchers' reflexivity – acknowledging and addressing potential biases or preconceptions that may have influenced the research process and outcomes.

-Providing information on the time allocation for the theoretical and practical workshops would offer insights into the intensity of participant engagement and the depth of knowledge transfer.

Result:

-Clarifying the total number of participants considered for each percentage would enhance transparency.

-While direct participant quotes are included, providing a more extensive context or explanation for selected quotes would help in interpreting the nuances of participant responses.

-Given the qualitative nature of the study and the use of Reflexive Thematic Analysis, it would be beneficial to include a brief reflection on the researchers' reflexivity and its potential impact on data interpretation.

Discussion:

-The discussion could benefit from a more detailed exploration of specific challenges encountered during the implementation of the MAF tool, providing insights into practical issues and potential solutions.

-While participants emphasize the importance of government actions, more specific policy recommendations or considerations for policymakers could be discussed. This would add depth to the suggestions for promoting broad access and minimizing potential risks.

-While the authors mention that the MAF tool is perceived as an innovation, they do not mention what makes it innovative. Needs more explanation.

Reviewer #2: Good Idea.

Some Grammatical mistakes to be corrected and spelling mistake in line 584 and some.

Data Analysis can be depicted in graph or pie chart for each component as it is a qualitative study.

Limitations and challenges faced during the study can be added.

Sample size is small and its a Pilot study and equal no of samples were not taken.

Reviewer #3: This study was aimed to the perceptions of professionals, caregivers, children, and 49 adolescents with disabilities regarding the implementation of the My Abilities First 50 (MAF) tool in Specialized Child Rehabilitation Centers (CERs). Authors concluded that The implementation of the MAF tool was 66 perceived as an innovation due to its focus on the abilities of individuals with 67 disabilities. However, there is a need to restructure it to broaden its scope and access 68 to different contexts in order to confront barriers and enhance the inclusion and 69 participation of children and adolescents with disabilities. Overall, the study is interesting, however there are some clarifications needed.

Comment#1

The authors should provide information about sample size estimation in this study.

Comment#2

The authors should provide appropriate information about the psychometrics properties of the “My Abilities First Tool”.

Comment#3

The authors should state limitations of this study.

6. PLOS authors have the option to publish the peer review history of their article (what does this mean?). If published, this will include your full peer review and any attached files.

Reviewer #1: No

Reviewer #2: No

Reviewer #3: No

---

## [Author Response · Author response to Decision Letter 0]

7 Mar 2024

Renato S. Melo, PhD

Academic Editor

PLOS ONE

On behalf of all the authors, I would like to express my gratitude for the opportunity to revise and resubmit the manuscript [PONE-D-23-41828], entitled 'Implementation of the My Abilities First Tool: a qualitative study on the perceptions of professionals, caregivers, children, and adolescents with disabilities', for publication in PLOS ONE. We greatly appreciate and value the time and attention you and the reviewers dedicated to providing feedback on our work. We are particularly grateful for the insightful comments that have led to significant improvements. We have meticulously incorporated the suggestions provided by you and the reviewers.

At this time, we are confident that our responses sufficiently address the comments provided, and we are prepared to finalize the revised version of the manuscript, taking into consideration any additional suggestions that the reviewers may offer. We have attached all responses pertaining to the suggestions for the resubmission.

Below we have provided our specific point-by-point responses to the editor's and reviewers' comments and new suggestions, highlighted in blue. All proposed modifications and corrections have been accepted by all authors and incorporated into the manuscript to elevate its standard and quality.

We hereby certify that all authors have thoroughly reviewed the manuscript, confirm its originality, and affirm that it is neither under consideration nor published, either wholly or partially, in another journal or similar publication. All authors unanimously consent to the submission of this article to PLOS ONE.

Please do not hesitate to contact me as the corresponding author.

Roselene Ferreira de Alencar

Department of Physical Therapy. 

Federal University of Rio Grande do Norte,

Natal, Rio Grande do Norte, Brazil

E-mail: roselene.alencar@ufrn.br

---

## [Decision Letter · Decision Letter 1]

22 Mar 2024

Implementation of the My Abilities First Tool: a qualitative study on the perceptions of professionals, caregivers, children, and adolescents with disabilities

PONE-D-23-41828R1

Dear Dr. Alencar,

We’re pleased to inform you that your manuscript has been judged scientifically suitable for publication and will be formally accepted for publication once it meets all outstanding technical requirements.

Kind regards,

Renato S. Melo, PhD

Academic Editor

PLOS ONE

Additional Editor Comments (optional):

Reviewers' comments:

Reviewer's Responses to Questions

**Comments to the Author**

1. If the authors have adequately addressed your comments raised in a previous round of review and you feel that this manuscript is now acceptable for publication, you may indicate that here to bypass the “Comments to the Author” section, enter your conflict of interest statement in the “Confidential to Editor” section, and submit your "Accept" recommendation.

Reviewer #1: All comments have been addressed

Reviewer #3: All comments have been addressed

2. Is the manuscript technically sound, and do the data support the conclusions?

Reviewer #1: Yes

Reviewer #3: Yes

3. Has the statistical analysis been performed appropriately and rigorously? 

Reviewer #1: Yes

Reviewer #3: Yes

4. Have the authors made all data underlying the findings in their manuscript fully available?

Reviewer #1: Yes

Reviewer #3: Yes

5. Is the manuscript presented in an intelligible fashion and written in standard English?

Reviewer #1: Yes

Reviewer #3: Yes

6. Review Comments to the Author

Reviewer #1: (No Response)

Reviewer #3: (No Response)

7. PLOS authors have the option to publish the peer review history of their article (what does this mean?). If published, this will include your full peer review and any attached files.

Reviewer #1: No

Reviewer #3: No

---

## [Editor Report · Acceptance letter]

2 May 2024

PONE-D-23-41828R1 

PLOS ONE

Dear Dr. Alencar, 

I'm pleased to inform you that your manuscript has been deemed suitable for publication in PLOS ONE. Congratulations! Your manuscript is now being handed over to our production team.

Kind regards, 

on behalf of

Dr. Renato S. Melo 

Academic Editor

PLOS ONE